# Synergistic Inhibitory Effect of Berberine and Low-Temperature Plasma on Non-Small-Cell Lung Cancer Cells via PI3K-AKT-Driven Signaling Axis

**DOI:** 10.3390/molecules28237797

**Published:** 2023-11-27

**Authors:** Tingting Lu, Yu Wang, Fang Liu, Lu Zhang, Simin Huang, Yuanyuan Zhou, Hui Wu, Yanmei Mao, Chufeng Jin, Wencheng Song

**Affiliations:** 1Key Laboratory for the Application and Transformation of Traditional Chinese Medicine in the Prevention and Treatment of Major Pulmonary Diseases, Anhui University of Chinese Medicine, Hefei 230012, China; tingting.lu@ahtcm.edu.cn (T.L.); wy1774086564@163.com (Y.W.); zhanglu626262@163.com (L.Z.); jasmine56521@163.com (S.H.); maoyanmei199507@ahtcm.edu.cn (Y.M.); 2Anhui Province Key Laboratory of Medical Physics, Institute of Health & Medical Technology, Hefei Institutes of Physical Science, Chinese Academy of Sciences, Hefei 230031, China; 17853728872@163.com (F.L.); yyz920507590@163.com (Y.Z.); wh18326114800@126.com (H.W.); 3Key Laboratory of Xin’an Medicine, Ministry of Education, Anhui University of Chinese Medicine, Hefei 230012, China; 4Key Laboratory of Neutronics and Radiation Safety, Chinese Academy of Sciences, Hefei 230031, China; 5International Academy of Neutron Science, Qingdao 266199, China

**Keywords:** low-temperature plasma (LTP), berberine (BER), non-small-cell lung cancer, sensitivity, PI3K-AKT signaling pathway

## Abstract

Low-temperature plasma (LTP) is an emerging biomedical technique that has been proposed as a potential approach for cancer therapy. Meanwhile, berberine (BER), an active ingredient extracted from various medical herbs, such as Coptischinesis, has been proven antitumor effects in a broad spectrum of cancer cells. In this study, we seek to develop a novel dual cancer therapeutic method by integrating pre-administration of BER and LTP exposure and evaluating its comprehensive antitumor effect on the human non-small-cell lung cancer (NSCLC) cell lines (A549 and H1299) in vitro. Cell viability, cell cycle, cell apoptosis, and intracellular and extracellular ROS were investigated. The results showed that cotreatment of BER and LTP significantly decreased the cell viability, arrested the cell cycle in the S phase, promoted cell apoptosis, and increased intracellular and extracellular ROS. Additionally, RNA Sequencing (RNA-Seq) technology was used to explore potential mechanisms. The differentially expressed genes among different treatment groups of NSCLC cells were analyzed and were mainly enriched in the phosphatidylinositol-3-kinase (PI3K)/protein kinase B (AKT) signaling pathway. Moreover, cotreatment of BER and LTP notably depressed the total protein expression level of PI3K and AKT with immunoblotting. In conclusion, BER and LTP have a synergistic inhibitory effect on NSCLC cells via the PI3K-AKT signaling pathway, which could provide a promising strategy for supplementary therapy in the anti-NSCLC battle.

## 1. Introduction

Non-small-cell lung cancer (NSCLC) is relatively insidious, with strong invasion, recurrence, and metastasis, and it represents the major subtype and accounts for approximately 85% of lung cancer patients [1]. Despite the remarkable evolution of conventional therapies that are routinely being applied for NSCLC treatment in recent years [2,3,4], the problem of limitation persists [2]. Apparently, novel alternative approaches for NSCLC treatment should be explored to support current treatment strategies [5]. Low-temperature plasma (LTP) treatment has been proposed as a potential approach for cancer therapy for killing cancer cells selectively via the generation of reactive oxygen species (ROS) and reactive nitrogen species (RNS) [6,7] with minimal adverse effects on normal cells [8,9]. The biological effects of LTP are mainly caused by the joint action of various active components produced in the plasma discharge process [10]. LTP treatment showed significant antitumor ability in about 20 types of cancer cells in vitro [11,12,13,14,15,16] and also in tumors in vivo [16]. Nowadays, several LTP devices have been used in the clinic for cancerous lesions treatment [17]. 

Meanwhile, berberine (BER), an isoquinoline alkaloid found in Berberis Vulgaris or Coptis chinensis Franch., has been intensively investigated for a wide range of therapeutic applications, such as anti-inflammatory, antidiabetic, antibacterial, and antitumor effects [18]. Currently, BER has a wide spectrum of antitumor activities against several high-risk cancers, including colon cancer, gastric cancer, breast cancer, and leukemia [19,20], which has been shown to enhance the efficacy of radiotherapy and chemotherapy [21]. BER is a potential candidate as a relatively safe and promising adjuvant therapy to cure lung cancer, both alone and in combination with several conventional cancer therapies [22]. The latest studies reported that BER displayed an antiproliferative effect against NSCLC cells in vitro and in vivo, leading to the prolonged survival of tumor-bearing mice [23]. 

With the expanding actuation duration of the LTP exposure, the activity of cancer cells weakens in a time-dependent manner [11,24]. Notably, different cancer cells have different sensitivities to LTP treatment due to their genetic background [25,26,27]. While the lung is an organ with limited radiation exposure, long-term radiation therapy can lead to radiation tolerance [28]. Therefore, using drugs to increase the radiation sensitivity of lung cancer cells, especially low-dose radiation sensitivity, is extremely important for improving the efficacy of radiation therapy in NSCLC. The main goal of the present study is to develop a novel dual cancer therapeutic strategy by integrating BER and LTP and assess its underlying mechanism for NSCLC treatment in vitro. 

## 2. Results

### 2.1. Inhibitory Activity of BER and/or LTP against NSCLC Cells

We examined the antiproliferative effects of BER, LTP, and BER combined with LTP on NSCLC cells. The MTT assay showed that A549 cells were less sensitive to 30 μM BER for 24 h than H1299 cells (Figure 1a). With the prolongation of LTP exposure, the viability of A549 and H1299 cells lowered in a slightly time-dependent manner at 24 h after the LTP exposures. However, the inhibitory effect of LTP exposures for 15 s, 30 s, and 45 s on the cell viability of both cells was weak (Figure 1b). We selected different concentrations of BER pre-administration on A549 and H1299 cells for 24 h and then LTP exposures for 15 s, 30 s, and 45 s. The results indicated that LTP exposures significantly inhibited cell growth and promoted cell death in a time-dependent manner when the cells were pretreated with 20 μM BER for 24 h (Figure 1b). The viability of A549 and H1299 cells decreased to 69.94 ± 1.99, 61.27 ± 1.61, 52.32 ± 2.61%, 77.40 ± 1.44, 72.53 ± 1.70, and 51.25 ± 1.71% of the control after LTP treatment for 15 s, 30 s, and 45 s, respectively.

### 2.2. BER and/or LTP Induced Cell Cycle and Cell Apoptosis in NSCLC Cells

First, we examined the effects of BER, LTP, and BER combined with LTP on cell cycle progression with flow cytometry. The control group was neither treated with BER nor LTP. As expected, 20 μM BER and 15-s LTP exposure markedly blocked the cell cycle progression at the S phase in A549 and H1299 cells (Figure 2 and Appendix A). No obvious changes in cell cycle distribution had been tested with the BER treatment compared with the control. The percentage of the two NSCLC cells in the G0/G1 phase decreased to the valley with the LTP treatment for 15 s. A significantly increased percentage of S-phase cells was observed in the 20 μM BER and 15-s LTP treatment group (A549: ** *p* < 0.01 vs. control, H1299: **** *p* < 0.0001 vs. control). The G2/M-phase cells initially increased and reached the peak with the 15-s LTP treatment. 

Next, we detected the effects of BER, LTP, and BER combined with LTP on cell apoptosis with flow cytometry (Annexin V-FITC/PI staining). The 20 μM BER and 15-s LTP exposure induced both early apoptosis and late apoptosis in H1299 cells more remarkably compared with A549 cells (Figure 3 and Appendix A). However, there was no obvious change in the percentage of necrosis A549 cells with BER and/or LTP treatment. Moreover, integrating pre-administration of 20 μM BER and 15-s LTP exposure induced a significant increase in the total apoptosis in H1299 cells compared with the single use of LTP treatment for 15 s, nearly 3.81-fold. 

### 2.3. Intracellular and Extracellular ROS Levels Increased after BER and/or LTP Treatment

LTP irradiation could cause the accumulation of ROS in cells, thus affecting cell viability, cell cycle [29], and cell apoptosis [30,31]. After the A549 and H1299 cells were treated with 20 μM BER and/or LTP exposure, the intracellular ROS level was determined (fluorescence intensity represented intracellular ROS intensity). The concentration of ROS in A549 cells treated with the single use of LTP (15 s, 30 s, and 45 s) had no obvious change in trend after 24 h incubation. After pretreatment with BER overnight and LTP exposure for 15 s, 30 s, and 45 s, there was a significant increase in the intracellular ROS concentration in A549 cells compared with the control group after LTP treatment for 15 s, 30 s, and 45 s, nearly 1.92, and 3.57, respectively (Figure 4a). Also, as shown in Figure 4, the concentration of ROS in H1299 cells increased with LTP exposure time. When the LTP exposure time increased to 45 s, the ROS level significantly increased nearly 2.45-fold compared with the control group. Moreover, similar patterns were also observed in the A549 cells pretreated with 20 μM BER overnight and with LTP exposure for 15 s, 30 s, and 45 s, a significant increase in the intracellular ROS concentration compared with control group after LTP treatment for 15 s, 30 s, and 45 s, nearly 2.18, 1.76, and 1.51-fold, respectively (Figure 4a). Integrating the pre-administration of BER and LTP induced a distinct increase in ROS production in A549 and H1299 cells. 

To determine whether liquid phase active substances were triggered in BER and/or LTP–inhibited cell proliferation, we detected the concentration of H_2_O_2_ generated by BER and/or LTP treatment in the cell culture medium of A549 and H1299 cells. The results indicate that the content of H_2_O_2_ increased with the BER and/or LTP treatment processing. As shown in Figure 4b, After LTP treatment for 0 s, 15 s, 30 s, and 45 s, the concentration of H_2_O_2_ in A549 and H1299 cells was 6.31 ± 2.73, 61.13 ± 3.90, 66.53 ± 2.69, 69.79 ± 2.57, 12.53 ± 0.92, 60.39 ± 1.01, 67.93 ± 0.88, and 67.06 ± 1.03 μmol/L, respectively. After integrating pre-administration of 20 μM BER and LTP treatment for 15 s, 30 s, and 45 s, the concentration of H_2_O_2_ was 64.59 ± 2.74, 70.06 ± 2.49, 69.19 ± 2.53, 60.53 ± 1.85, 65.73 ± 2.10, and 65.33 ± 1.48 μmol/L, respectively. Additionally, the single use of BER also could produce a low concentration of H_2_O_2_ in the two NSCLC cells.

### 2.4. Transcriptome Sequencing and Assembly

There was a total of 12 samples each for A549 and H1299 cells samples, and the purity and integrity of total RNA in each sample fitted the requirements of database building and sequencing (Appendix A). A total of 18,513 genes were detected, with Q20% ≥ 96.83% and Q30% ≥ 91.98% in all samples, and there was little difference in clean reads among samples (Appendix A). After obtaining clean reads, we aligned clean reads to the reference genome sequence. The average comparison rate of each sample was 88.49%, and the uniform comparison rate between samples showed that the data between samples were comparable (Appendix A). Clean reads were compared to the reference gene sequence, and the average comparison rate of the sample comparison gene set was 70.54% (Appendix A). With calculating the Pearson correlation coefficient of all gene expressions between each two samples, the expression correlation between 12 samples of two cells was 0.8–0.9, indicating that the samples had good correlation and repeatability (Appendix A). PCA analysis results, gene expression levels distribution, expression volume, and gene expression of each sample are shown in Appendix A.

### 2.5. Functional Enrichment of Differentially Expressed Genes (DEGs)

Based on the preliminary experimental results, we found that A549 cells were less sensitive to combination therapy with BER and LTP compared to H1299 cells. To investigate the pathways related to the effect of integrating pre-administration of 20 μM BER and LTP exposure, we first conducted a differential analysis of sequencing data between H1299 cells that were relatively sensitive to LTP and A549 cells that were relatively tolerant to LTP. With padj < 0.05 and |log2FoldChange| > 1 as the threshold, we screened the differential genes and obtained 3292 high-expression genes and 3895 low-expression genes (Figure 5a and Appendix A). Then, the GO and KEGG enrichment analyses were carried out for these genes. We found that these genes were enriched in the PI3K-AKT signaling pathway [32,33,34], MAPK signaling pathway [35], ECM-receiver interaction [36,37], and other pathways (Figure 5a), and the influence on the biological processes was related to electromagnetic organization development and extracellular matrix. (Figure 5a). 

Next, we analyzed the difference between the BER group and BER combined with the LTP group in H1299 cells. The intersection of the high and low expression of the two difference results, 27 differentially up-regulated and 44 differentially down-regulated genes (Figure 5b) were obtained, and GO and KEGG enrichment analyses were also conducted. We found that these genes were also enriched in the PI3K-AKT signaling pathway [32,33,34], MAPK signaling pathway [32], ECM-receiver interaction [36,37], and other pathways, and the influence on the biological processes was mainly related to the Notch signaling pathway [38] and extracellular matrix (Figure 5b).

### 2.6. Protein–Protein Interaction (ppi) Analysis

We also used string database and cytoscape software to analyze the protein–protein interactions (ppi) of these common differential genes. These genes were divided into five separate networks. After the subnetworks were extracted with MCODE, four subnetworks (mcode1–4) were obtained. The PI3K-AKT signaling pathway [39,40] is a key link that modulates major mechanisms of radiation resistance: intrinsic radiosensitivity, tumor cell proliferation, and hypoxia. Therefore, we extracted six genes belonging to the PI3K-AKT signaling pathway from common differential genes, including PIK3CA, NGFR, PGF, NPNT, COL4A5, and COL4A4. There were significant differences in these genes between the BER group and BER combined with the LTP group, whether in A549 or H1299 cells (Appendix A). In addition, NGFR and PGF were in subnetwork 2, and COL4A5 and COL4A4 were in subnetwork 3, both of which were hub genes in the protein–protein interaction network.

### 2.7. Protein Expression Level Changed after BER and/or LTP Treatment

To confirm the key role of integrating pre-administration of BER and LTP exposure on the PI3K-AKT signaling pathway, we detected the total protein expression levels of PI3K and AKT of two NSCLC cells using protein immunoblotting (Figure 6). The synergistic inhibitory effect of BER and LTP potently decreased the expression levels of PI3K and AKT in H1299 cells. As expected, the two proteins’ expression by integrating 20 μM BER and LTP exposure for 30 s in A549 cells showed a slight downward trend and was not as obvious as the former. The results indicated that the combination treatment inhibited the expression levels of two major proteins on this pathway in two NSCLC cells and presented a trend of joint action that also supported the transcriptome analysis results.

## 3. Discussion

Although significant advances in the clinical treatment field of NSCLC have reduced the mortality rate of patients and improved their 5-year survival rate to some extent, multidrug resistance and treatment resistance remain key factors that hinder the long-term efficacy of clinical treatment. Due to the diversity and complexity of NSCLC cell types, as well as the consideration of disease stages and patients’ health status, the current trend of NSCLC in clinics gradually shifted from a focus on monotherapy to multimodal therapy [41]. A unique approach for multimodal therapy should be required for each tumor type due to the heterogeneity of the genetic background and the unique biological behavior. The efficacy and interaction of each therapy also dictate the selection of therapies for a multimodal approach [42]. The multimodal therapy results in remarkable “1 + 1 > 2” effects. This study further provided the basis for seeking new therapies for NSCLC in clinics.

In this study, the concentration of intracellular ROS in A549 cells treated with the single use of LTP (15 s, 30 s, and 45 s) had no obvious change in trend after 24 h incubation from fluorescence images. The results were not consistent with what the other authors reported. We believed that LTP exposure (15 s, 30 s, and 45 s) could induce intracellular ROS in A549 cells, but the contents were relatively low. This may be due to the different sensitivities of NSCLC cells to LTP. Moreover, the intracellular ROS were detected by a fluorescent probe DCFH-DA, which was qualitative. Given that cotreatment of BER and LTP has exerted the synergistic inhibitory effect in NSCLC cells, especially changing the characteristics of A549 cells, more mechanisms of combinative action may be worth exploring.

Elevated PI3K activity is well established as a hallmark of cancer, which is associated with diverse oncogenes and growth factor receptors. AKT is a critical signaling node, which regulates cellular functions that are closely related to tumorigenesis [40]. The PI3-K/AKT pathway plays a crucial role in intrinsic radiosensitivity, tumor cell proliferation, and hypoxia. Targeted inhibition of the PI3K/AKT signaling pathway may enhance tumor control in NSCLC, antagonize the radiation-induced cellular defense mechanisms, and increase radiosensitivity in tumors [40]. Interestingly, integrating pre-administration of BER and 15-s LTP exposure and the single use of LTP treatment for 15 s did not effectively decrease the expression levels of PI3K and AKT, showing a slight downward trend in A549 cells, which was not as obvious as the results in H1299 cells. This also may be due to the genetic background of the different NSCLC cells.

Although cotreatment of BER and LTP has shown a synergistic inhibitory effect on NSCLC cells in vitro, deeper in vivo research urgently needs to be carried out. In addition, the introduction of active ingredients of traditional Chinese medicine into the multimodal approach should be of high priority. Multimodal therapy initially had the potential as a candidate method for clinical treatment of NSCLC.

## 4. Materials and Methods

### 4.1. Cell Culture and Reagents

The human NSCLC cell lines A549 and H1299 were purchased from the Cell Bank of Type Culture Collection of Chinese Academy of Sciences (Shanghai, China). They were maintained in RPMI Medium 1640 medium (GIBCO, Shanghai, China) supplemented with 2% L-glutamine, 10% fetal bovine serum (LONSERA, Shanghai, China), and 1% penicillin/streptomycin (NCM Biotech, Suzhou, China). Cells were cultured in a humidified incubator (Thermo, Waltham, MA, USA) at 37 °C under 5% CO_2_. They were adherent cells, which were grown in tissue culture dishes to 85–95% confluence before use. 

BER was purchased from a commercial chemical vendor (Shanghai Jiuzhi Chemical, Shanghai, China). Stock solutions (30 mmol/L) were prepared in 100% dimethyl sulfoxide (DMSO). Gradient dilution is required according to the experiment.

### 4.2. Plasma Device and Cell Treatment

The dielectric barrier discharge (DBD) plasma device operates at a power of 24.40 W that is measured using an oscilloscope (DSOX2024A, Keysight Technologies, Sisha Rosa, CA, USA) through the Lissajous method (Figure 7). The LTP generator consists of a reactor chamber with two pairs of electrodes, one air inlet, and one outlet, as sufficiently described in our previous studies [43]. The high-voltage electrode is a 58 mm diameter copper plate covered by a layer of dielectric materials quartz medium (1 mm thick). The grounding electrode is a 62 mm diameter copper cylinder. The flow rate of Helium gas (99.99%) is 2.6 L/min. Helium gas is injected into the reactor cavity for 90 s to eliminate the residual air in the cavity.

Place the cell culture dish (60 mm) containing the sample on the grounding electrode and control the distance between the sample and quartz at 5 mm. Before each discharge, the working gas is delivered to the device for 90 s to remove the residual air in the device. For LTP exposures, cells are seeded into 60 mm diameter Petri dishes with 4 mL complete culture medium and exposed to LTP for 15 s, 30 s, and 45 s, incubated for a further 24 h, and then harvested for the next experiments. The control cells (LTP exposure for 0 s) are subjected to identical procedures except the LTP treatment. 

### 4.3. Cell Viability Assay

A549 and H1299 cells were grown in cell culture dishes (60 mm) for 12 h, and various concentrations of the BER were added to a portion of them for 24 h. Cell viability was determined using the MTT assay (Sigma-Aldrich, Burlington, MA, USA) after the BER treatment and/or LTP exposure. The cells were treated with 10% MTT (Sigma-Aldrich, USA) solution for 4 h at 37 °C, and then the supernatant was discarded. A total of 100 μL of MTT formazan solution in DMSO (Sangon Biotech, Shanghai, China) was transferred into the 96-well plate. The optical density (OD) values were determined at 492 nm by using an enzyme plate analyzer (Hiwell-Diatek, Wuxi, China). Data were normalized to the control groups (DMSO) and are presented as the mean of three independent measurements with a standard error of <10%. The values were calculated using GraphPad Prism 5.0 (GraphPad Software, San Diego, CA, USA) [44]. 

### 4.4. Cell cycle Analysis and Cell Apoptosis Detection

A549 and H1299 cells were treated with 20 μM BER treatment for 24 h and/or 15-s LTP exposure. The cells were fixed in 70% cold ethanol, incubated at −20 °C overnight, and stained with a PI/RNase staining buffer (Beyotime Biotechnology, Shanghai, China). The samples were examined using a FACS Calibur flow cytometer (BD, Franklin Lakes, NJ, USA), and the results were analyzed using ModFit LT v5.0.9 software. Cell apoptosis was also detected using the Annexin V-FITC apoptosis detection kit (Beyotime Biotechnology, Shanghai, China). The operation procedure was performed as previously described [45].

### 4.5. ROS Measurement

The intracellular ROS were detected with a fluorescent probe DCFH-DA (Beyotime Biotechnology, Shanghai, China) following the manufacturer’s instruction. After the 20 μM BER treatment and/or LTP exposure (15 s, 30 s, 45 s) or 5 mM ROS scavengers (N-acetyl-l-cysteine, NAC) treatment for 2 h, the culture was incubated at 37 °C for 6 h, and then was stained with DCFH-DA solution (10 µM) for 30 min at 37 °C in the dark. The supernatant was discarded and cleaned with phosphate-buffered saline (PBS) 3 times. The fluorescence was determined with a fluorescence microscope (Olympus, Tokyo, Japan).

The H_2_O_2_ level in the cell culture medium was determined with a hydrogen peroxide assay kit (Beyotime Biotechnology, Shanghai, China). After the 20 μM BER treatment and/or LTP exposures (15 s, 30 s, and 45 s), 50 μL medium was immediately added to the 96-well plate, and a 100 μL hydrogen peroxide detection reagent was added according to kit instructions. The absorbance at 560 nm was measured with an enzyme plate analyzer.

### 4.6. RNA Extraction, Library Preparation, and Sequencing

Total RNA was isolated from A549 and H1299 cells using a PureLink RNA purification kit (ThermoFisher). Then, rRNA was removed using the ribosomal rRNA removal Kit (Illumina, San Diego, CA, USA) for each sample. NEBNext Ultra II RNA Library Kit, NEBNext Multiplex Oligos, and Primer Set 1 (New England Biolabs, Ipswich, MA, USA) were used to construct the RNA library. The library was sequenced at 150 bp on Illumina HiSeq 2500.

### 4.7. Sequencing Data Processing

FastQC v0.11.7 (https://github.com/s-andrews/FastQC, accessed on 1 June 2022) and trim_galore v0.6.3 (https://github.com/FelixKrueger/TrimGalore, accessed on 1 June 2022) were used for sequencing raw data for quality control, and the software of STAR v2.7.10b (https://github.com/alexdobin/STAR, accessed on 3 June 2022) was performed on the obtained clean data for the genome comparison. The genome and annotation files were from the GENCODE (release 35) database. The software of RSEM v1.3.1 (https://github.com/de weylab/RSEM, accessed on 4 June 2022) was used for splicing, merging, and quantifying transcripts, and then the transcriptome count expression profile data was obtained.

### 4.8. Gene Expression Analysis and Enrichment Analysis

The R language (v4.1.1) DESeq2 package (v1.34.0) to standardize and analyze the transcriptome count data was applied to explore the synergistic inhibitory mechanism of BER and LTP on NSCLC cells. The obtained differential genes were enriched with GO and KEGG using the clusterProfiler package (v4.6.2), and the string database (https://cn.string-db.org/, accessed on 9 June 2022) and MCODE plug-in of cycloscape software (v3.7.2) were used for the analysis of protein interaction (PPI) and hub gene. The gene sets of KEGG, HALLMARK, BIOCARTA, and REACTOME were downloaded from the msigdb database (https://www.gsea-msigdb.org/gsea/msi gdb/index.jsp, accessed on 9 June 2022), and a GSEA analysis was conducted for the results of difference in each group.

### 4.9. Signaling Pathway Studies 

A549 and H1299 cells were treated as 2.4 and washed with cold 1 × PBS. Whole-cell protein was extracted with RIPA lysis buffer (Beyotime Biotechnology, Shanghai, China) at 4 °C for 30 min. The protein concentration was determined with a BCA protein assay kit (Beyotime Biotechnology, Shanghai, China). Western blotting was performed as previously described [30]. Equivalent amounts of protein samples were separated on sodium dodecyl sulfate-polyacrylamide gel electrophoresis (SDS-PAGE gel) and transferred onto a polyvinyl-lidene difluoride membrane. The following antibodies were used to detect specific proteins and should be within the concentration range indicated by the manufacturer: phosphatidylinositol-3-kinase (PI3K) (#3011) and protein kinase B (AKT) (#4685) were from Cell Signalling Technology (CST). The GAPDH antibody (#HC301-02) was purchased from TransGen Biotechnology.

### 4.10. Statistical Analysis

Data were presented as the Mean ± SEM of three independent experiments. Statistical significance was calculated using a *t*-test to compare each treatment group to the corresponding vehicle control, and significance was indicated by asterisks: * *p* < 0.05, ** *p* < 0.01, *** *p* < 0.001, **** *p* < 0.0001.

## 5. Conclusions

In summary, the cotreatment of BER and LTP can significantly inhibit the NSCLC cell viability, arrest the NSCLC cell cycle in the S phase, induce the NSCLC cell apoptosis, and increase intracellular and extracellular ROS in vitro. BER and LTP have a synergistic inhibitory effect on NSCLC cells via the PI3K-AKT signaling pathway, which may lead to a shift in the paradigm of NSCLC therapy.

## Figures and Tables

**Figure 1 molecules-28-07797-f001:**
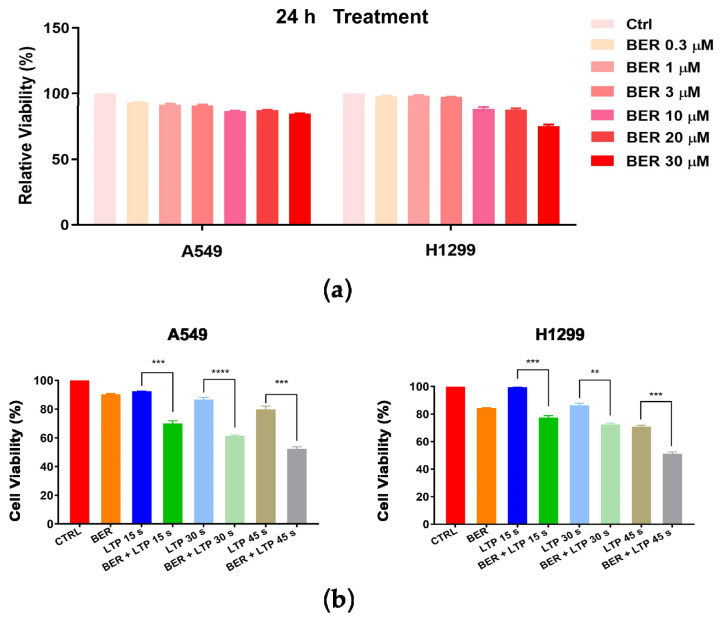
Effects of BER and/or LTP on cell viability (MTT assay). (**a**) A549 and H1299 cells were pretreated with different concentrations of BER (0.3 μM, 1 μM, 3 μM, 10 μM, 20 μM, 30 μM). (**b**) A549 and H1299 cells were treated with 20 μM BER and/or LTP for 15 s, 30 s, and 45 s. At 24 h after LTP treatment, cell viability was measured with the MTT assay. Data were presented as the Mean ± SEM of triplicate tests and significance was indicated by asterisks: ** *p* < 0.01, *** *p* < 0.001, **** *p* < 0.0001.

**Figure 2 molecules-28-07797-f002:**
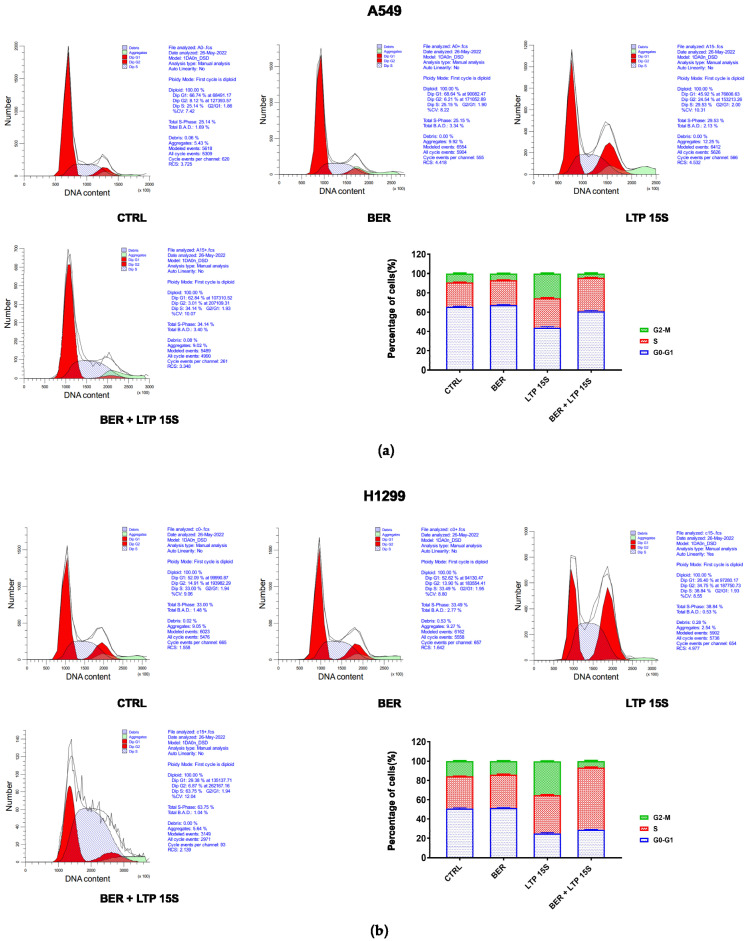
Effects of BER and/or LTP on cell cycle progression. (**a**) Histograms and statistical analysis of cell cycle in A549 cells with 20 μM BER and/or 15-s LTP exposure (flow cytometry). (**b**) Histograms and statistical analysis of cell cycle in H1299 with 20 μM BER and/or 15-s LTP exposure (flow cytometry). Data were presented as the Mean ± SEM of three independent experiments.

**Figure 3 molecules-28-07797-f003:**
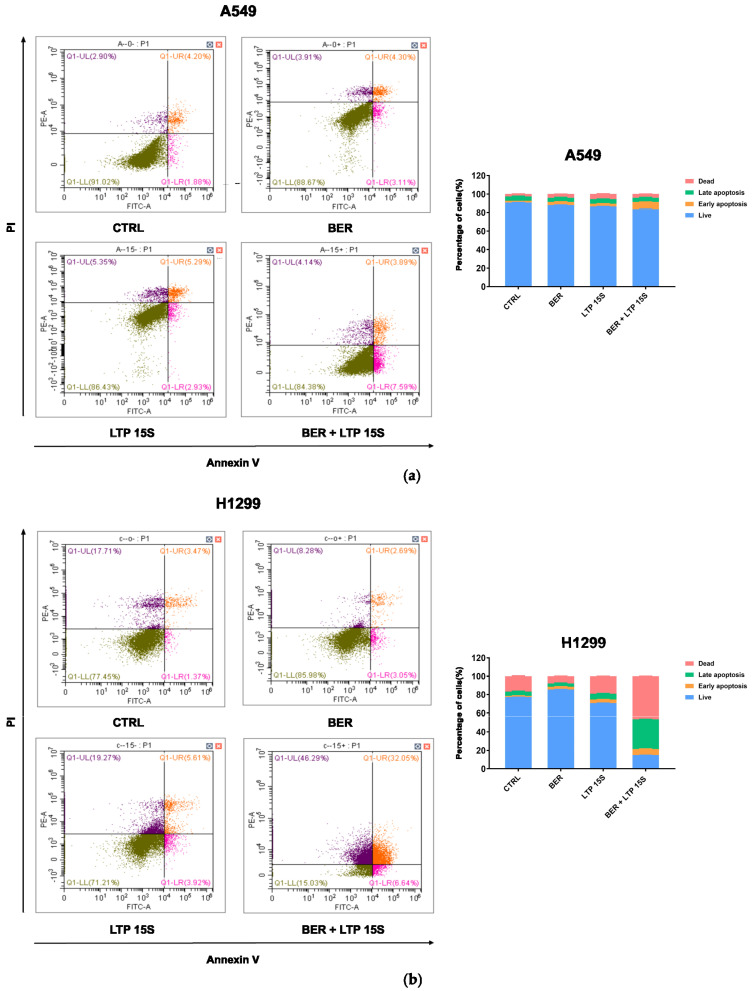
Effects of BER and/or LTP on cell apoptosis progression. (**a**) Dot plots and statistical analysis of cell apoptosis in A549 cells with 20 μM BER and/or 15−s LTP exposure (flow cytometry). (**b**) Dot plots and statistical analysis of cell apoptosis in H1299 cells with 20 μM BER and/or 15−s LTP exposure (flow cytometry). Q1−UL: (Annexin V−FITC)−/PI+, cells in this area were necrotic cells. Q1−UR: (Annexin V−FITC) +/PI+, cells in this region were late apoptotic cells. Q1−LR: (Annexin V−FITC)+/PI−, cells in this region were early apoptotic cells. Q1−LL: (Annexin V−FITC)−/PI−, cells in this region were living cells. (**b**) Statistical analysis of cell-apoptosis analysis of A549 and H1299 cells with 20 μM BER and/or 15−s LTP exposure. Data were presented as the Mean ± SEM of three independent experiments.

**Figure 4 molecules-28-07797-f004:**
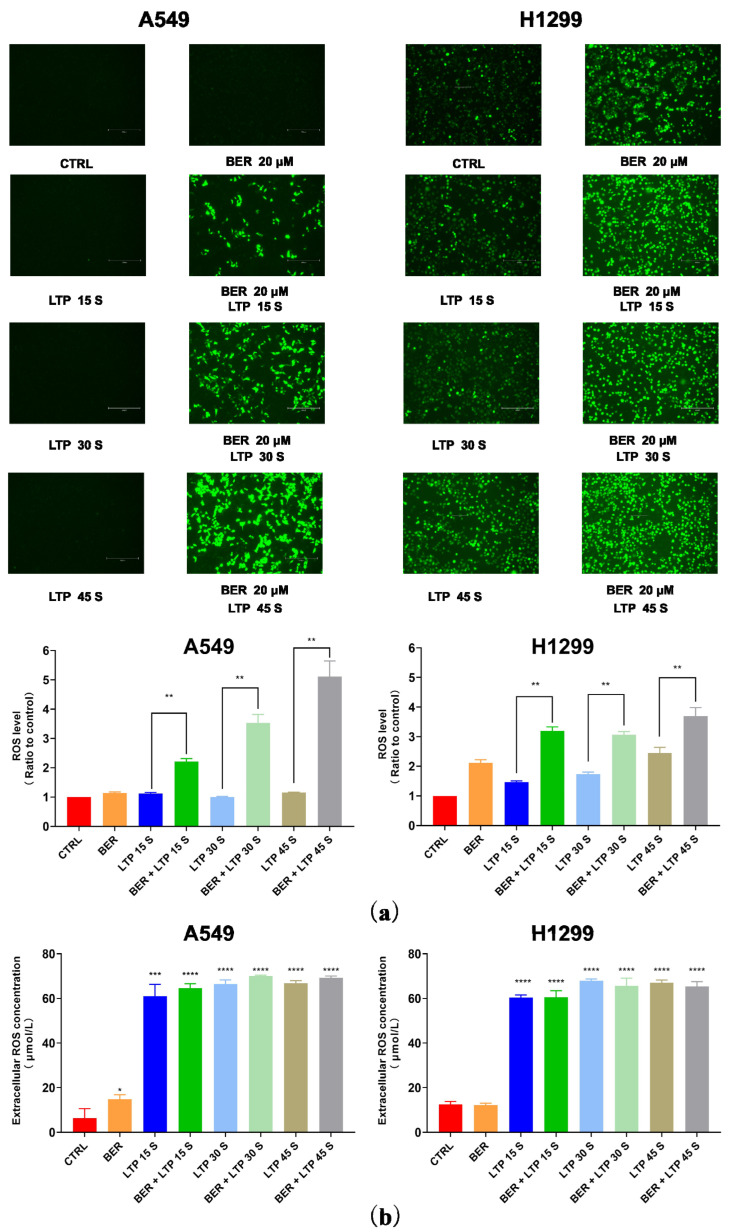
Intracellular ROS levels and extracellular H_2_O_2_ content. (**a**) Fluorescence images of intracellular ROS generation in A549 and H1299 cells. The quantification by measuring fluorescence pixel intensity using Image J software (v1.47). (**b**) The concentrations of H_2_O_2_ in A549 and H1299 cell culture mediums. The data represented the Mean ± SEM of three independent experiments. The significance was indicated by asterisks: * *p* < 0.05, ** *p* < 0.01, *** *p* < 0.001, **** *p* < 0.0001.

**Figure 5 molecules-28-07797-f005:**
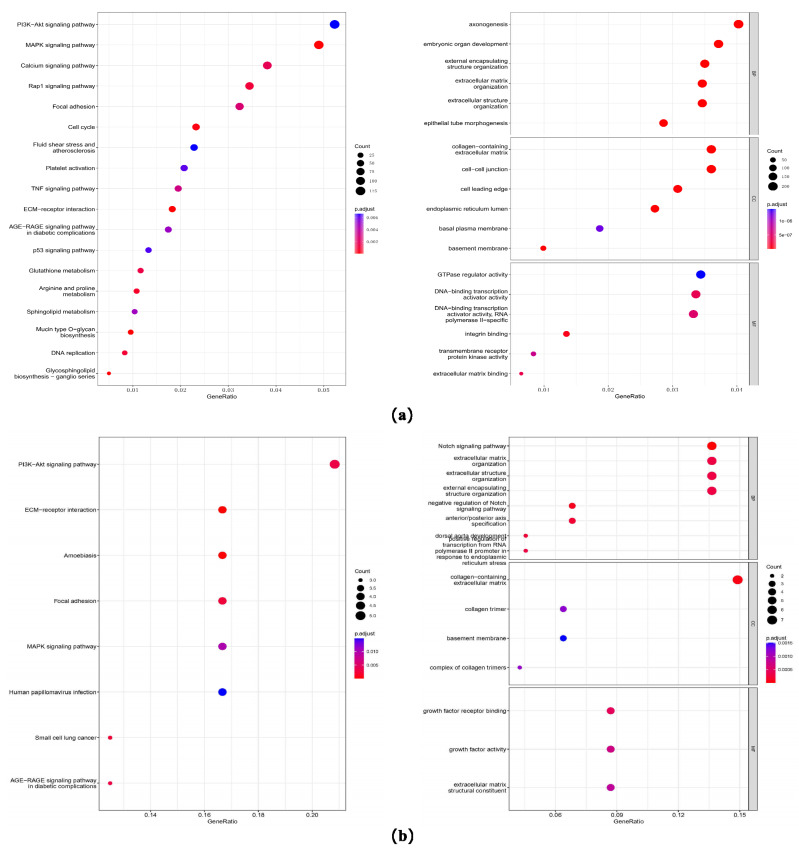
Gene expression and GO/KEGG enrichment analysis on the sequencing data. (**a**) Dot plots of GO and KEGG enrichment analysis of differential genes of H1299 cells that are relatively sensitive to LTP and A549 cells that are relatively tolerant to LTP. The abscissa is the percentage of differential genes in this pathway, the ordinate is the name of the pathway, the color of the dot represents the *p*-value, and the size represents the number of differential genes. (**b**) Dot plots of GO and KEGG enrichment analysis of differential genes on the sequencing data of the BER group and BER combined with the LTP group in LTP-sensitive H1299 cells. The abscissa is the percentage of differential genes in this pathway, the ordinate is the name of the pathway, the color of the dot represents the *p*-value, and the size represents the number of differential genes.

**Figure 6 molecules-28-07797-f006:**
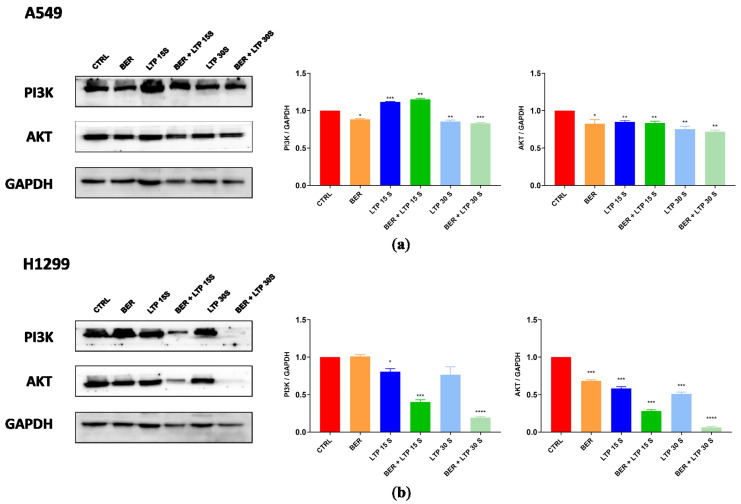
BER and/or LTP treatment induced changes in protein expression levels (immunoblotting). (**a**) BER and/or LTP treatment affected the expression of PI3K and AKT in A549 cells. (**b**) BER and/or LTP treatment affected the expression of PI3K and AKT in H1299 cells. The significance was indicated by asterisks: * *p* < 0.05, ** *p* < 0.01, *** *p* < 0.001, **** *p* < 0.0001.

**Figure 7 molecules-28-07797-f007:**
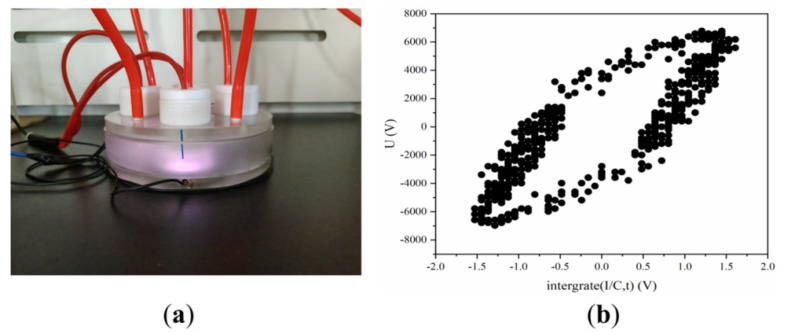
DBD LTP apparatus. (**a**) Image of pictures of the LTP apparatus. (**b**) Lissajous pattern of DBD.

## Data Availability

Data are contained within the article and Appendix A.

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
