# Peer review of "Synergistic Inhibitory Effect of Berberine and Low-Temperature Plasma on Non-Small-Cell Lung Cancer Cells via PI3K-AKT-Driven Signaling Axis"

_molecules, 2023, doi:10.3390/molecules28237797_

Round 1
Reviewer 1 Report
Comments and Suggestions for Authors
The manuscript describes a study that could provide useful data for development of novel dual cancer therapeutic method by combination of administration of Berberine (chemical extracted from herbs) and low-temperature plasma exposure. Authors studied and evaluated the effect of such approach on the in-vitro cell model of lung cancer cells and investigated viability, cell cycle, cell apoptosis and intracellular and extracellular ROS changes. The study has original approach and investigations of this type can provide useful information related to applications of plasma in cancer treatment. Although the research is relevant for the field, the manuscript needs improvement and revision before publication. Details are given below.
2. Results
Lines 76-80 and also in other places in the text: stating results with errors should contain error rounded to one exceptionally two significant digits and the result value should be rounded accordingly.
Lines 91-92: 'The control group did not take any intervention measure' - sentence is not clear, please change and explain
Lines 92-103: please describe in more details results presented on Fig.2 a) and b). Describe different labels shown in Fig.2b)
Fig.2b) text given with the plots is not visible. Please change this.
Fig.3a) plot numbers on axis are too small to be readable, please change this.
Lines 127-129: many other author reported that the treatment of A549 cell using low-temperature plasma induces effect in ROS at different time scales but also after 24h. Please comment in Discussion part here why the effect was missing in this study.
Lines 129-132: sentence not clear, please rephrase
Please add either few sentences in Discussion part or make Conclusion part highlighting particular achievements, main quantitative data and information obtained in this study.
Comments on the Quality of English LanguageQuality of English is good, few sentences need to be rephrased in order to be precise.
Author Response
Reviewer #1:
The manuscript describes a study that could provide useful data for development of novel dual cancer therapeutic method by combination of administration of Berberine (chemical extracted from herbs) and low-temperature plasma exposure. Authors studied and evaluated the effect of such approach on the in-vitro cell model of lung cancer cells and investigated viability, cell cycle, cell apoptosis and intracellular and extracellular ROS changes. The study has original approach and investigations of this type can provide useful information related to applications of plasma in cancer treatment. Although the research is relevant for the field, the manuscript needs improvement and revision before publication. Details are given below.
- Lines 76-80 and also in other places in the text: stating results with errors should contain error rounded to one exceptionally two significant digits and the result value should be rounded accordingly.
Reply: According to the reviewer’s suggestions, we have looked through the whole manuscript again and corrected results with errors as one exceptionally two significant digits (52.32 ± 2.61%, 72.53 ± 1.70, 4.41-fold).
- Lines 91-92: 'The control group did not take any intervention measure' - sentence is not clear, please change and explain.
Reply: According to the reviewer’s suggestions, we have revised the sentence as: “The control group was neither treated with BER nor LTP.”
- Lines 92-103: please describe in more details results presented on Fig.2 a) and b). Describe different labels shown in Fig.2b)
Reply: According to the reviewer’s suggestions, we described in more details results presented on Fig.2 a) and b) as following in the manuscript. The Fig. 2 and figure legend have also been revised accordingly.
“As expected, 20 μM BER and 15-s LTP expose markedly blocked the cell cycle progression at the S phase in A549 and H1299 cells (Figure 2, Supplementary Fig. 1). No obvious changes of cell cycle distribution had been tested with the BER treatment compared with control. The percentage of the two NSCLC cells in G0/G1 phase decreased to the valley with the LTP treatment for 15 s. A significantly increased percentage of S-phase cells were observed in 20 μM BER and 15-s LTP treatment group (A549: **p < 0.01 vs control, H1299: ****p < 0.0001 vs control). G2/M-phase cells initially increased and reached the peak with 15-s LTP treatment.”
- Fig.2b) text given with the plots is not visible. Please change this.
Reply: Thanks for reviewer’s kindly comment and suggestion, which is very important for us to improve our manuscript. We have changed the layout of Fig 2 and corrected the legend as “(a) Histograms and statistical analysis of cell-cycle in A549 cells with 20 μM BER and/or 15-s LTP expose (flow cytometry). (b) Histograms and statistical analysis of cell-cycle in H1299 with 20 μM BER and/or 15-s LTP expose (flow cytometry)” to make it more clear to the readers. In addition, we have tried to separate the G0/G1, S, G2/M phase and calculated the p-value. This was listed in the Supplemental Fig. 1.
- Fig.3a) plot numbers on axis are too small to be readable, please change this.
Reply: Thanks for reviewer’s kindly comment and suggestion, we have changed Fig.3 to make it be readable to the readers. The figure legend of Fig 3 has also been revised it accordingly.
“(a) Dot-plots and statistical analysis of cell-apoptosis in A549 cells with 20 μM BER and/or 15-s LTP expose (flow cytometry). (b) Dot-plots and statistical analysis of cell-apoptosis in H1299 cells with 20 μM BER and/or 15-s LTP expose (flow cytometry)”
- Lines 127-129: many other author reported that the treatment of A549 cell using low-temperature plasma induces effect in ROS at different time scales but also after 24h. Please comment in Discussion part here why the effect was missing in this study.
Reply: Thanks for reviewer’s kindly comment and suggestion. We have commented in Discussion part.
“In this study, the concentration of intracellular ROS in A549 cells treated with the single use of LTP (15 s, 30 s and 45 s) had no obvious change in trend after 24 h incubation from fluorescence images. The results were not consistent with other authors reported. We believed that LTP exposes (15 s, 30 s and 45 s) could induce intracellular ROS in A549 cells, but the contents were relatively low. These may be due to the different sensitivities of NSCLC cells to LTP. Moreover, the intracellular ROS were detected by a fluorescent probe DCFH-DA, which was qualitative. Given that co-treatment of BER and LTP have exerted the synergistic inhibitory effect in NSCLC cells, especially changing the characteristics of A549 cells, more mechanisms of combinative action may be worth exploring.
- Lines 129-132: sentence not clear, please rephrase
Reply: Thanks for reviewer’s kindly comment and suggestion, we have revised the text.
“Moreover, similar patterns were also observed in the A549 cells with pre-treated 20 μM BER overnight and LTP exposure for 15 s, 30 s and 45 s, a significant increase in the intracellular ROS concentration compared with control group after LTP treatment for 15 s, 30 s and 45 s, nearly 2.18, 1.76 and 1.51-fold, respectively (Figure 4a)”.
- Please add either few sentences in Discussion part or make Conclusion part highlighting particular achievements, main quantitative data and information obtained in this study.
Reply: Thanks for reviewer’s kindly comment and suggestion, we have added a concrete conclusion on particular achievements, main quantitative data and information of this study in section 5 of the manuscript.
“In summary, the co-treatment of BER and LTP can significantly inhibit the NSCLC cells viability, arrest the NSCLC cells cycle in S phase, induce the NSCLC cells apoptosis and increase intracellular and extracellular ROS in vitro. BER and LTP have a synergistic inhibitory effect on NSCLC cells via PI3K-AKT signaling pathway, which may lead to a shift in the paradigm of NSCLC therapy”.
Reviewer 2 Report
Comments and Suggestions for Authors
I suggest that the manuscript be returned to the authors for thorough revision.
The following points need to be addressed:
1. Abstract - It should be written on which cell lines the experiments were performed (full names, and abbreviations)
2. Introduction - It is necessary to refer to the results of recent research as well as an overview of recently published reviews on the effects of berberine on tumors and tumor cell models (e.g.: Goel A. Chem Biol Drug Des. 2023 Jul;102(1):177-200. doi: 10.1111/cbdd.14231; Clin Exp Pharmacol Physiol . 2022 Jan;49(1):134-144. doi: 10.1111/1440-1681.13582; Achi IT, Sarbadhikary P, George BP, Abrahamse H. Cells. 2022 Oct 31;11(21):3433. doi: 10.3390/cells11213433).
3. Results - Results are not adequately described for all experiments performed. Figure 2 (a) is not legible. The image description is unclear.
4. The discussion in the manuscript is too general. The results obtained have not been compared with the results of previous studies nor adequately discussed in the Discussion chapter.
5. A concrete conclusion should be written on the basis of the obtained results and a further direction of research should be proposed.
Comments on the Quality of English Language
English syntax must be improved.
Author Response
Reviewer #2:
I suggest that the manuscript be returned to the authors for thorough revision. The following points need to be addressed:
- Abstract - It should be written on which cell lines the experiments were performed (full names, and abbreviations)
Reply: Thanks for reviewer’s kindly comment and suggestion, we have added the cell lines which were performed in experiments and described as following in the manuscript.
“In this study, we seek to develop a novel dual cancer therapeutic method by integrating pre-administration of BER and LTP expose, and evaluate its comprehensive anti-tumour effect on the human non-small cell lung cancer (NSCLC) cell lines (A549 and H1299) in vitro”.
- Introduction - It is necessary to refer to the results of recent research as well as an overview of recently published reviews on the effects of berberine on tumors and tumor cell models (e.g.: Goel A. Chem Biol Drug Des. 2023 Jul;102(1):177-200. doi: 10.1111/cbdd.14231; Clin Exp Pharmacol Physiol . 2022 Jan;49(1):134-144. doi: 10.1111/1440-1681.13582; Achi IT, Sarbadhikary P, George BP, Abrahamse H. Cells. 2022 Oct 31;11(21):3433. doi: 10.3390/cells11213433).
Reply:Thanks for the reviewer’s valuable suggestion. We have added the results of recent research on the effects of berberine on tumours and tumour cell models and described as following in the manuscript. The new references ([24] [25] [26]) have also been updated accordingly.
“Currently, BER has a wide spectrum of anti-tumour activities against several high-risk cancers, including colon cancer, gastric cancer, breast cancer, and leukemia [22, 23], which has been shown to enhance the efficacy of radiotherapy and chemotherapy [24]. BER is a potential candidate as a relatively safe and promising adjuvant therapy to cure lung cancer, alone and in combination with several conventional cancer therapies [25]. Latest studies reported that BER displayed the anti-proliferative effect against NSCLC cells in vitro and in vivo, leading to prolonged survival of tumour-bearing mice [26]”.
- Results - Results are not adequately described for all experiments performed. Figure 2 (a) is not legible. The image description is unclear.
Reply: Thanks for reviewer’s kindly comment and suggestion. First, we have described in more details results presented on Fig.2 a) and b) as following in the manuscript; We aslo changed the layout of Fig 2 and corrected the legend to make it more clear to the readers. Second, we have described in more details results presented on Fig.3 as following in the manuscript, and have changed the layout of Fig 3 and corrected the legend.
“As expected, 20 μM BER and 15-s LTP expose markedly blocked the cell cycle progres-sion at the S phase in A549 and H1299 cells (Figure 2, Supplementary Fig. 1). No obvious changes cell cycle distribution had been tested with the BER treatment compared with control. The percentage of the two NSCLC cells in G0/G1 phase decreased to the valley with the LTP treatment for 15 s. A significantly increased percentage of S-phase cells were observed in 20 μM BER and 15-s LTP treatment group (A549: **p < 0.01 vs control, H1299: ****p < 0.0001 vs control). G2/M-phase cells initially increased and reached the peak with 15-s LTP treatment”.
“(a) Histograms and statistical analysis of cell-cycle in A549 cells with 20 μM BER and/or 15-s LTP expose (flow cytometry). (b) Histograms and statistical analysis of cell-cycle in H1299 with 20 μM BER and/or 15-s LTP expose (flow cytometry)”
“20 μM BER and 15-s LTP expose induced both early apoptosis and late apoptosis in H1299 cells more remarkably compared with A549 cells (Figure 3, Supplementary Fig. 2). However, there was no obvious change of the percentage in necrosis A549 cells with BER and/or LTP treatment. Moreover, integrating pre-administration of 20 μM BER and 15-s LTP expose induced a significantly increase of the total apoptosis in H1299 cells compared with the single use of LTP treatment for 15 s, nearly 3.81-fold”.
“(a) Dot-plots and statistical analysis of cell-apoptosis in A549 cells with 20 μM BER and/or 15-s LTP expose (flow cytometry). (b) Dot-plots and statistical analysis of cell-apoptosis in H1299 cells with 20 μM BER and/or 15-s LTP expose (flow cytometry)”
- The discussion in the manuscript is too general. The results obtained have not been compared with the results of previous studies nor adequately discussed in the Discussion chapter.
Reply: Thanks reviewer for the suggestion, we have revised and added some of the discussions.
“In this study, the concentration of intracellular ROS in A549 cells treated with the single use of LTP (15 s, 30 s and 45 s) had no obvious change in trend after 24 h incubation from fluorescence images. The results were not consistent with other authors reported. We believed that LTP exposes (15 s, 30 s and 45 s) could induce intracellular ROS in A549 cells, but the contents were relatively low. These may be due to the different sensitivities of NSCLC cells to LTP. Moreover, the intracellular ROS were detected by a fluorescent probe DCFH-DA, which was qualitative. Given the fact that co-treatment of BER and LTP have exerted the synergistic inhibitory effect in NSCLC cells, especially changing the characteristics of A549 cells, more mechanisms of combinative action may be worth exploring”.
- A concrete conclusion should be written on the basis of the obtained results and a further direction of research should be proposed.
Reply: Thanks reviewer for the suggestion. We have written a concrete conclusion on the basis of the obtained results and a further direction in section 5 of the manuscript.
“In summary, the co-treatment of BER and LTP could significantly inhibit the NSCLC cells viability, arrest the NSCLC cells cycle in S phase, induce the NSCLC cells apoptosis and increase intracellular and extracellular ROS in vitro. BER and LTP have a synergistic inhibitory effect on NSCLC cells via PI3K-AKT signaling pathway, which may lead to a shift in the paradigm of NSCLC therapy”.
Round 2
Reviewer 1 Report
Comments and Suggestions for Authors
The authors made changes according to the comments and questions in the review and therefore I propose to publish the manuscript in the present form.
Reviewer 2 Report
Comments and Suggestions for Authors
Dear Editor,
the manuscript has been considerably improved.
Comments on the Quality of English Language Minor editing of English language required.